# Is the Numbness after Knee Replacement a More Critical Complication Than Thought? A Detailed Analysis of Neuropathic Pain and Functional Outcomes

**DOI:** 10.3390/medicina58101369

**Published:** 2022-09-29

**Authors:** Deniz Gurler, Ismail Buyukceran

**Affiliations:** 1Department of Orthopedics and Traumatology, University of Health Sciences, Samsun Training and Research Hospital, Samsun 55090, Turkey; 2Department of Orthopedics and Traumatology, Faculty of Medicine, Ondokuz Mayis University, Samsun 55139, Turkey

**Keywords:** knee replacement, neuropathic pain, complications, knee joint innervation, numbness

## Abstract

*Background and Objectives*: Numbness, one of the complications after knee replacement (KR), has been studied far less than other complications, and there are few studies on this topic. Without comprehensive research on numbness, there is a need to design a study that includes all relevant parameters. This study investigated the relationship between numbness and pain and neuropathic pain and its impact on functional and emotional life and the functioning of the knee. *Materials and Methods*: The 105 knees with KRs were divided anteriorly into six regions. Numbness was determined with a pinprick test. Douleur Neuropathique 4 (DN4) and a painDETECT questionnaire (PD-Q) were tested for neuropathic pain. Patients’ physical, emotional, and social status and functional knee abilities were assessed with the Short Form (SF-36) and the Western Ontario and McMaster Universities Osteoarthritis Index (WOMAC) test. The relationships between numbness and gender, age, bilaterality, postoperative follow-up time, and incision measures were examined. The binomial logistic regression model was applied to investigate the effects of age, gender, bilaterality, incision length, and postoperative follow-up time on numbness. *Results*: A total of 88.6% of the patients were female, half had bilateral KRs, and the mean age was 68.3 years. Numbness occurred more frequently in the L3 and L2 areas than in other regions. There was no association with gender, bilaterality, and age, but there was a weak association with postoperative follow-up time. There was a positive correlation between numbness and neuropathic pain. It has been found that there is a significant relationship between numbness and physical function, and it has a negative effect. Emotional health was not significantly associated with numbness. The effect of numbness on social functioning was statistically significant. Knee functioning was measured with the WOMAC, and significant differences were found in the numbness group. *Conclusions*: According to the results, numbness is considered to be a complication that affects social and physical life and has a negative impact on functional outcomes of the knee. The results show that it should be considered a statistically significant complication. Numbness and its indirect effects should be considered in patients whose follow-up parameters are normal over one year but whose symptoms do not resolve.

## 1. Introduction

Knee replacement (KR) is used worldwide as the first option for treating patients with end-stage gonarthrosis. About 106,000 KRs are performed every year in Turkey [1]. We know these numbers have decreased during the pandemic, but 2030 projections say that this procedure will increase by 600% [2]. There will likely be complications in such a trendy operation. The most common complications are anterior knee pain, stiffness, implant infection, deep venous thrombosis, and septic or aseptic loosening. Especially among the complications of knee prostheses, deep infection, which is both an economic and clinical problem at a rate of up to 12%, is more common and has been frequently studied [3]. Numbness is not considered as significant a complication as it should be. Different data on incidence have been published, ranging from 37–55% [4] to 71–100% [5]. Even in terms of incidence, the wide range of results and uncertain, conflicting data indicate that the topic deserves a new study.

The main objective of this study is to determine the impact of numbness on neuropathic pain, non-neuropathic pain, physical function, social function, emotional health status, and functional outcomes of the knee after KR.

## 2. Materials and Methods

The study was approved on 1 December 2021 by the Ethics Committee in the Samsun Training and Research Hospital with decision number GOKA-2021-19-2. The study was prepared prospectively. The criteria for patients who participated in this series were as follows: KR must have occurred at least one year ago. The same surgeon should have performed the midline skin incision and medial parapatellar approach, and the posterior ligament protection implant should have been used with a mobile insert. Special care was taken to ensure that all study participants performed the same type of incision and technique (medial parapatellar) to standardize the effects of the incision. The fact that different incisions were not included in the study made the group comparable. There should have been no other postoperative complications, no history of rheumatologic treatment, no diabetic neuropathy, no cerebrovascular disease, and no spinal surgery. All patients should have received spinal anesthesia during surgery, not peripheral nerve block.

During the examination, the knee region of the patients was divided into six areas. These were the medial and lateral sides with the suprapatellar area, the patellar area, and the infrapatellar area. These areas were numbered 1, 2, and 3 laterally and medially (suprapatellar L1-M1, patellar L2-M2, infrapatellar L3-M3). Participants were asked in which areas they felt numbness. Only those areas that could be confirmed with the pinprick test, needle, and brush were recorded. Douleur Neuropathique 4 (DN4) [6], painDETECT questionnaire (PD-Q) [7], Short Form SF-36 (PF: Physical Function, RLPH: Role Limit of Physical Health, RLEP: Role Limit of Emotional Problem EW: Emotional Well-being, Energy, SF: Social Function, Pain, GH: General Health, HC: Health Change) [8] and Western Ontario and McMaster Universities Osteoarthritis Index (WOMAC) (Pain, Stiffness, Physical Function, Total) [9] scales were completed. The scores of the SF-36 subgroups were calculated online [10]. Patients’ neuropathic pain was assessed with DN4 and PD-Q scales. Emotional status, physical and social functioning, and knee functioning were measured with SF-36 and WOMAC. Patients were admitted to the Orthopedic Outpatient Clinic of Samsun Training and Research Hospital between 15 November 2021, and 27 April 2022. Three hundred fifty-two patients were studied in the outpatient clinic. One hundred and fifteen patients met the study criteria. Seventy-eight patients agreed to participate in the study. Data were collected from 70 patients who fully completed the study form. A single clinician (DG) examined all patients in the study. Thirty-five of the patients had bilateral KR. Each knee was recorded as individual data. Two separate groups were formed for patients with and without postoperative numbness. Participants without numbness were subjected to statistical analysis as a control group for the effects of numbness.

### Statistical Analysis

Patient data were recorded in a database. The database was transferred to Jamovi 2.3.2.0 (the Jamovi Project, Sydney, Australia) [11] software. Descriptive statistics and frequencies of demographic data (gender, age, bilaterality, postoperative follow-up time, and incision size) were generated. Chi-square correlation (χ^2^-test for association) was used for nominal variables (gender, bilaterality). Spearman’s correlation (Rho: ρ) was used for continuous variables (age, postoperative follow-up time, incision size). Mann–Whitney U test was used for non-normally distributed data (between numbness, neuropathic pain, non-neuropathic pain, social function, physical function, general health, and knee function).

The severity of neuropathic pain was measured using the DN4 and PD-Q scales. SF-36 (Pain) and WOMAC (Pain) data were analyzed for the relationship between pain and numbness. SF-36 (Role Limit of Physical Health, Physical Function) and WOMAC (Function, Stiffness) data were interpreted for the relationship between physical function and numbness. SF-36 data (Role Limit of Emotional Problem, Emotional Well-Being) were evaluated for the effect of numbness on emotional health. Social function, general health, and functional knee capacity were evaluated with SF-36 (Social Function), SF-36 (General Health), and WOMAC (total) scores. A binomial logistic regression model was applied to measure the influence of variables (gender, age, bilaterality, postoperative follow-up time, and incision size) on numbness at the scar site. Ordinal logistic regression analysis was performed to determine the relationship between the six knee areas with other parameters (pain, neuropathic pain, physical function, emotional health, and functional knee capacity).

## 3. Results

A total of 93/105 (88.6%) female and 12/105 (11.4%) male knees were included in the study. Half of the participants were bilateral, 35/70 (50%). The mean age was 68.3 ± 7.35 years, and the postoperative follow-up time was 40.3 ± 35.8 months. The size of the incisional scar was 14.3 ± 1.69 cm. Of the patients with numbness, 33/105 (31.4%) were affected. The distribution of numbness by location: L1 12/105 (11.4%), L2 27/105 (25.7%), L3 30/105 (28.6%), M1 0/105 (0%), M2 5/105 (4.8%), M3 11/105 (10.5%) (Table 1). From the data in the table, L3 5/105 (4.8%), L2 2/105 (1.9%), L2 + L3 13/105 (12.4%), L1 + L2 + L3 12/105 (11.4%), M3 6/105 (5.7%), and M2 + M3 5/105 (4.8%) stand out (Figure 1) [12].

Ordinal logistic regression was used to determine whether there were differences between the six individual regions, lateral and medial regions, and finally, between lateral and medial regions. When L1-L2-L3-M2-M3 (M1 was removed because the value is zero) are examined together, none of the localizations are related to the parameters (neuropathic pain, pain, physical function, emotional health). L1-L2-L3 have a significant relationship between DN4 and L3 (neuropathic pain). M2-M3 has no significant relationship with other parameters, but M3 was associated with PD-Q. When comparing L and M, there is a significant relationship between DN4 and PD-Q and L (β = 2.905, *p* < 0.001, β = 1.102 *p* = 0.009).

Neuropathic pain scales were calculated as DN4 5.17 ± 1.99 (*p* = 0.003) and PD-Q 18.2 ± 6.6 (*p* = 0.009). SF-36 subgroup data (*p* < 0.001) are shown in Table 2. The data of the WOMAC subgroup (*p* < 0.001) are shown in Table 3.

Numbness was not associated with gender (χ^2^ = 0.260, *p* = 0.610), bilaterality (χ^2^ = 0.795, *p* = 0.672), and length of incision (ρ = −0.082, *p* = 0.408). Postoperative follow-up time (ρ = −0.386, *p* < 0.001) and age (ρ = −0.202, *p* = 0.039) were weakly and negatively associated with numbness.

The reliability of SF-36 subgroups (α = 0.864) and WOMAC subgroups (α = 0.804) was high. With numbness, SF-36 (pain) scores (U = 744, *p* = 0.002) and WOMAC (pain) scores (U = 893, *p* = 0.040) were higher than without numbness. Pain scores were found to be significantly higher in patients with numbness.

No significant association was found between SF-36 (Role Limit of Physical Health) (U = 1136, *p* = 0.700) and WOMAC (Stiffness) (U = 9 63, *p* = 0.110) and numbness. SF-36 (Physical Function) (U = 870, *p* = 0.027) and WOMAC (Physical Function) (U = 864, *p* = 0.025) were significantly associated with numbness. It can be said that numbness after KR significantly affects the patient’s overall physical level, especially since the two main scores used in the study showed a significant correlation with the physical function data.

It was found that patients’ emotional health was not associated with numbness: SF-36 (Role Limit of Emotional Problem) (U = 1092, *p* = 0.463) and WOMAC (Emotional Well Being) (U = 1022, *p* = 0.244). Since the two data on emotional state do not show any significant relationship, we can clearly say that the emotional state of the patients is not affected by numbness.

The effect on SF-36 (social function) was statistically significant (U = 831, *p* = 0.011). There was no significant association with SF-36 (general health) (U = 1010, *p* = 0.206). Knee functional capacity was measured with the WOMAC, and significant differences were found in the numbness group (U = 862, *p* = 0.024). According to these results, it was found that numbness after KR did not affect the patient’s general health but negatively influenced social life and knee functioning.

A model was constructed to predict numbness with demographic variables (age, bilaterality, incision, postoperative follow-up time) (Model 1). Because the numbness variable was dichotomous, the model was constructed using binomial logistic regression. The results of Model 1 were statistically significant (pseudo R^2^ = 0.181, χ^2^ = 23.7, *p* < 0.001). A significant difference in postoperative follow-up time was found (β = −0.05086, *p* = 0.004). According to the possible ratios between all model variables, an odds ratio of 0.950 was found for the follow-up time (Table 4). Statistically, no predictive variable was found in the model.

No correlation was found between numbness and gender, age and bilaterality, incision size, emotional status, and general health. There was a correlation between neuropathic pain and follow-up time and a high correlation with pain, physical function, and functional capacity of the knee.

## 4. Discussion

The oldest work on the innervation of the knee dates back to the work of Ellis in 1840. The first modern medical study was the study by Gardner in 1948 [13]. According to the results of these studies, sensory innervation of the anterior knee is provided by eight nerves Figure 2.

Our study used the innervation of the six regions as a reference for sensory studies. L1 lateral branch of nerve to vastus intermedius (LBNVM), L2 nerve to vastus lateralis (NVL), L3 inferolateral genicular nerve (ILGN) and recurrent fibular nerve (RFN), M3 inferomedial geniculate nerve (IMGN) and infrapatellar branch of saphenous nerve (IPBSN), M2 superomedial geniculate nerve (SMGN), and M1 nerve to vastus medialis (NVM). However, the IPBSN responsible for numbness in the lateral region has been the most studied in the papers. In the 2013 study by Kerver et al., the anatomical variations and sensory distribution were examined in detail, and it was found that iatrogenic injuries were common in knee surgeries [14]. In the 2012 study by Leliveld et al., IPBSN injury was found to cause anterior knee pain after tibial nailing [15]. It is known that the location and length of incisions used in knee surgery have significant effects on numbness [16,17,18,19].

However, in our study, no significant correlation was found between the length of the incision scar and numbness sensation. This result can be attributed to choosing a minimally invasive method as the preferred incision in our patients. When the anterior knee is divided into six regions, an average numbness area of 2.56 units can be calculated if each region is considered a unit. If this unit calculation is proportional to the total knee region, it can be deduced that an average of 42.7% of the knee region is affected by numbness. It means that an extensive area is affected. It was observed that the lateral flap was not affected in only one of the patients with numbness. This result suggests that nerve injury is more common in the lateral area. However, numerous medial numbness lesions were also observed (11/33, 33.3%). In a 2017 study by Jariwala et al., although it has been noted that females have more frequent problems with numbness, we did not find such an association in our study. [20]. No significant difference was found in the relationship between numbness, which has never been investigated in previous studies, and whether they had bilateral surgery in the same session. A significant relationship with the length of follow-up of patients in the study was assumed to the patients’ selection. The patients had to undergo KR at least one year before. The fact that some of the numbness around the knee improved within one year and that there was no improvement after one year in previous studies supports our findings [4,21].

Neuropathic pain was noted to varying degrees in 40% of patients using the PD-Q scale and in 50% of patients due to the DN4 scale. Previous studies have published that the rate of neuropathic pain after KR is 40.4% [22]. The results of this study are consistent with the results of previous studies. Numbness correlated with PD-Q and moderately with DN4 from these two scales. It may be significant because some postoperative chronic knee and neuropathic pain are associated with numbness around the knee. Therefore, numbness is more important than it looks.

Non-neuropathic pain is also a fundamental problem after KR. In their 2011 study, Wylde et al. found that the rate of resistant pain after KR was 15% [23]. The study assessed the pain with WOMAC (Pain) and SF-36 (Pain). The high correlation between both scales and numbness indicates that the numbness group has more severe pain problems.

In studies that examined the relationship between numbness and physical function in patients, the ability to kneel and move was usually examined [4,24]. However, this study found no significant association between SF-36 (RLPH) and WOMAC (Stiffness). However, there was a significant association between SF-36 (Physical Function) and WOMAC (Physical Function) and numbness. Thus, it can be concluded that numbness negatively impacts physical function significantly.

In the studies by Borley et al. and Tsudaka et al., the rate of numbness affecting life was 7–8.7%. Although it was impossible to determine a rate in our study, SF-36 (RLEP) and the WOMAC correlation of well-being were examined to investigate the emotional state of patients. No significant correlation with numbness was found. In previous studies, it was found that the emotional state of patients was not affected by numbness. In our study, we came to results that confirmed this. SF-36 Social Function did not show a significant relationship also supports these findings. Similarly, it is essential to note that numbness was not significantly associated with SF-36 (General Health) status. Numbness does not affect emotional state, social life, or general health status.

In the 2017 study by Blackburn et al., no association was found between numbness and WOMAC scores [5]. On the contrary, our study found a significantly positive relationship between WOMAC score and numbness. The decreased functional capacity of the knee in patients with numbness was interpreted as statistically significant.

The major limitation of the study is the lack of preoperative control data. Another limitation of the study is that the measurements were not blinded. We could get more scientific and meaningful results if they were available. Nevertheless, the results of the study contain essential information. In addition, the number of patients could be higher to obtain more precise statistical results.

## 5. Conclusions

When patients with KRs have numbness that persists for at least a year, the numbness, although partially associated with neuropathic pain, affects their daily social and physical life and knee function without affecting their emotional health. These results show that numbness is an indirect factor affecting patient satisfaction. The presence of this complication should be questioned, especially in patients who are radiologically unremarkable but have chronic pain and dysfunction.

Numbness after knee replacement is, therefore, a significant complication. Orthopedic surgeons should consider this and try to prevent numbness from indirectly satisfying the patient.

## Figures and Tables

**Figure 1 medicina-58-01369-f001:**
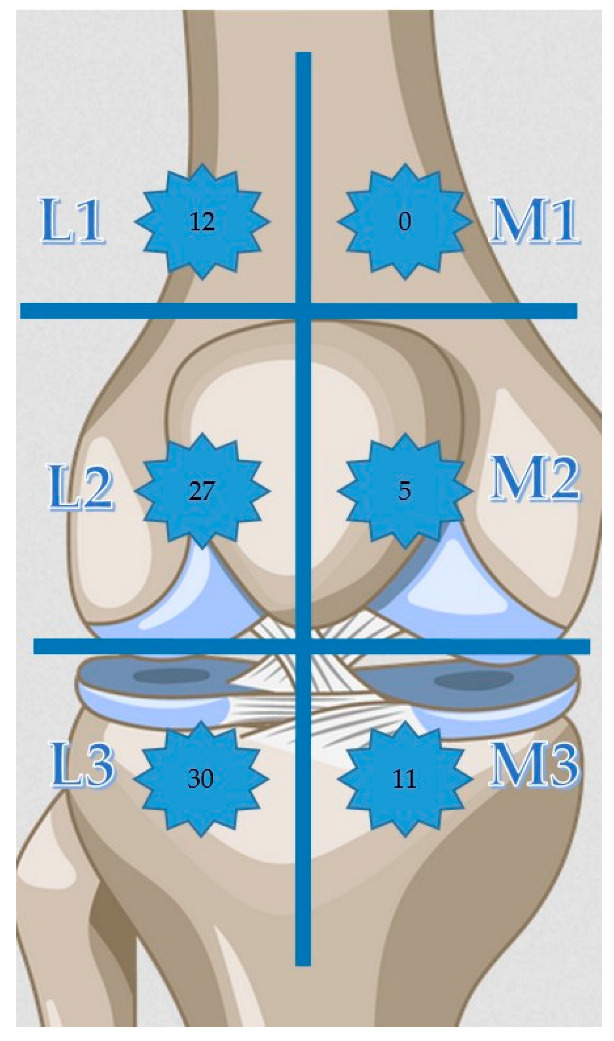
Distribution of numbness. Note L: lateral, M: medial, suprapatellar L1-M1, patellar L2-M2, infrapatellar L3-M3.

**Figure 2 medicina-58-01369-f002:**
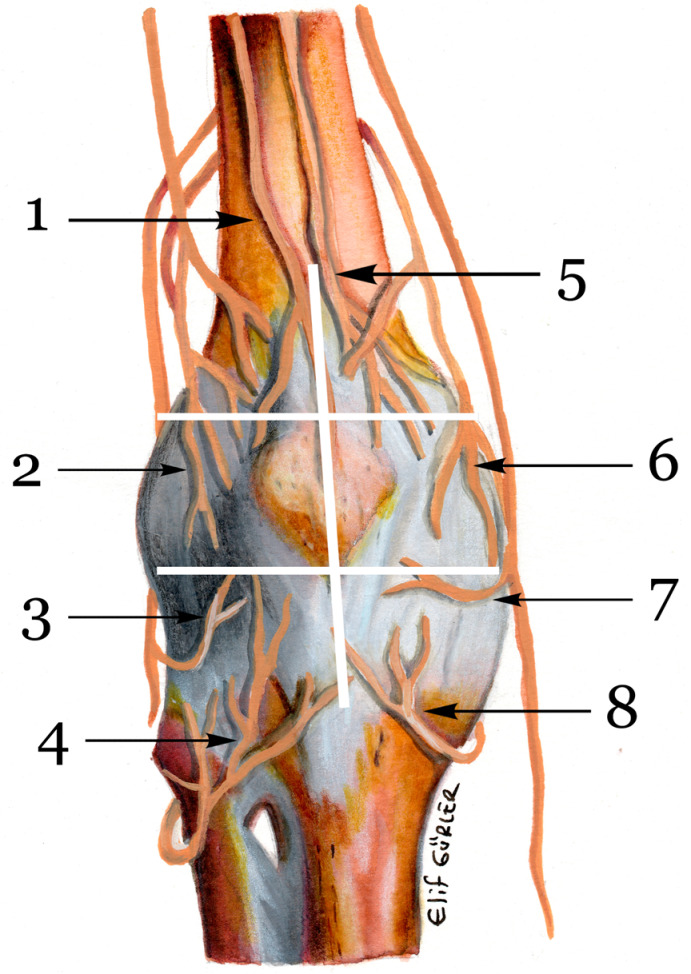
Sensorineural anatomy of knee. Note 1. lateral branch of nerve to vastus intermedius (LBNVM), 2. nerve to vastus lateralis (NVL), 3. inferolateral genicular nerve (ILGN), 4. recurrent fibular nerve (RFN), 5. nerve to vastus medialis (NVM), 6. superomedial geniculate nerve (SMGN), 7. infrapatellar branch of saphenous nerve (IPBSN), 8. inferomedial geniculate nerve (IMGN).

**Table 1 medicina-58-01369-t001:** Distribution of the numbness to the skin areas around the incision.

*n*	L1	L2	L3	M1	M2	M3
1	-	+	-	-	-	-
2	-	+	+	-	-	+
3	-	+	-	-	-	-
4	-	+	+	-	-	-
5	-	+	+	-	-	-
6	-	+	+	-	-	-
7	+	+	+	-	-	-
8	+	+	+	-	-	-
9	+	+	+	-	-	-
10	+	+	+	-	-	-
11	-	-	+	-	-	-
12	-	+	+	-	-	-
13	-	+	+	-	-	-
14	+	+	+	-	-	-
15	+	+	+	-	-	-
16	-	+	+	-	-	-
17	+	+	+	-	-	-
18	-	+	+	-	-	-
19	+	+	+	-	+	+
20	+	+	+	-	-	-
21	-	+	+	-	-	-
22	+	+	+	-	+	+
23	-	+	+	-	-	-
24	-	+	+	-	-	-
25	+	+	+	-	-	+
26	-	-	+	-	-	+
27	-	-	+	-	-	+
28	-	+	+	-	-	+
29	-	-	+	-	+	+
30	-	+	+	-	-	+
31	-	-	+	-	+	+
32	-	-	-	-	+	+
33	+	+	+	-	-	-

Note. L: lateral, M: medial, suprapatellar L1-M1, patellar L2-M2, infrapatellar L3-M3. +: numbness, -: no numbness.

**Table 2 medicina-58-01369-t002:** SF-36 Descriptive statistics of subgroups of Short Form 36 (SF-36).

	PF	RLPH	RLEP	Energy	EW	SF	Pain	GH	HC
*N*	105	105	105	105	105	105	105	105	105
Mean	69.7	60.7	62.9	50.7	53.7	68.4	67.3	51.5	65.2
Median	75	75	66.7	55	56.0	75.0	67.5	50	75
Standard deviation	18.8	20.5	25.5	8.55	6.51	18.4	22.5	6.54	21.8
Minimum	10	0	0.00	25	32.0	12.5	10.0	35	25
Maximum	95	100	100	70	68.0	100	100	70	100
Skewness	−1.36	−0.516	−0.466	−0.635	−0.736	−0.243	−0.493	−0.251	−0.210
Std. error skewness	0.236	0.236	0.236	0.236	0.236	0.236	0.236	0.236	0.236
Kurtosis	1.44	0.406	0.135	0.530	1.05	0.210	−0.362	0.230	−0.578
Std. error kurtosis	0.467	0.467	0.467	0.467	0.467	0.467	0.467	0.467	0.467
Shapiro–Wilk W	0.858	0.856	0.833	0.927	0.934	0.918	0.934	0.941	0.872
Shapiro–Wilk *p*	<0.001	<0.001	<0.001	<0.001	<0.001	<0.001	<0.001	<0.001	<0.001

Note. PF: Physical Function, RLPH: Role Limit of Physical Health, RLEP: Role Limit of Emotional Problem EW: Emotional Well-Being, SF: Social Function, GH: General Health, HC: Health Change.

**Table 3 medicina-58-01369-t003:** WOMAC Knee Functional Scale Subgroups.

	Wpain	Wstiff	WPFunc	WOMAC
*N*	105	105	105	105
Mean	5.77	2.27	20.0	29.2
Median	5	2	18	25.0
Standard deviation	3.81	1.61	12.8	18.6
Minimum	1	0	4	5.21
Maximum	15	6	51	72.9
Skewness	0.618	0.461	0.622	0.636
Std. error skewness	0.236	0.236	0.236	0.236
Kurtosis	−0.588	−0.741	−0.476	−0.483
Std. error kurtosis	0.467	0.467	0.467	0.467
Shapiro–Wilk W	0.915	0.907	0.923	0.926
Shapiro–Wilk *p*	< 0.001	<0 .001	< 0.001	<0 .001

Note. Wpain: WOMAC Pain (0–20), Wstiff: WOMAC Stiffness (0–8), WPFunc: WOMAC Physical Function (0–68), WOMAC: total Score (0–100).

**Table 4 medicina-58-01369-t004:** Model 1 coefficients.

Predictor	Estimate (β)	SE	Z	*p*	Odds Ratio
POFT	0.0517	0.0173	2.987	0.003	1.053
Age	0.0102	0.0343	0.297	0.766	1.010
INS	−0.0150	0.1360	−0.110	0.912	0.985
BL	0.0346	0.2928	0.118	0.906	1.035

Note. Estimates represent the log odds of “Numbness = 0” vs. “Number = 1”. POFT: Postoperative Follow-up Time (month), INS: incision measure. BL: Bilaterality.

## Data Availability

The sharing link can access study statistics reports. https://www.dropbox.com/s/c6q3ws8a4jlxgma/numbness%20v13%20hepsi.rar?dl=0 (accessed on 16 August 2022).

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
