# Peer review of "Is the Numbness after Knee Replacement a More Critical Complication Than Thought? A Detailed Analysis of Neuropathic Pain and Functional Outcomes"

_medicina, 2022, doi:10.3390/medicina58101369_

Round 1

Reviewer 1 Report

The study title is focused on the numbness after knee replacement. This is a very interesting topic to search possibilities to solve patients problems after total knee replacement. There is erea that should be improved before publication

  1. Please explain why there is no control group in this study.
  2. Please describe what approach was used in total knee replacement and consider to describe approach impact on your topic
  3. Innervation area is different than distribution presented on figure 1. It can have impact on results especially when 2 regions are involved.
  4. Please rewrite conclusions and add your results. Conclusions from abstract should be similar to final

Reviewer 2 Report

The manuscript is interesting in regard to the topic addressed, with a not so small case history.

The metodological approach is correct and the conclusions are in line with the results obtained.

It could be worthy of publication, being very relevant to the important issue, however an extensive linguistic revision by a native English speaker or Editing system is a priority (there are some grammatical errors in the manuscript, the English used is not always scientifically correct).

It useful recommend the following citation: PMID: 33739013

Round 2

Reviewer 1 Report

Responses and corrections accepted. I recommend to accept in present form.